# The Impact of Lockdowns on Caffeine Consumption: A Systematic Review of the Evidence

**DOI:** 10.3390/ijerph19095255

**Published:** 2022-04-26

**Authors:** Dimitra Rafailia Bakaloudi, Kleo Evripidou, Ranil Jayawardena, João Breda, Theodoros Dardavessis, Kalliopi-Anna Poulia, Michail Chourdakis

**Affiliations:** 1Laboratory of Hygiene, Social & Preventive Medicine and Medical Statistics, School of Medicine, Faculty of Health Sciences, Aristotle University of Thessaloniki, 541 24 Thessaloniki, Greece; didibak3@gmail.com (D.R.B.); kleo_evripidou@hotmail.com (K.E.); dardaves@auth.gr (T.D.); 2Department of Physiology, Faculty of Medicine, University of Colombo, Colombo 00700, Sri Lanka; ranil@physiol.cmb.ac.lk; 3WHO European Office for the Prevention and Control of NCDs, WHO Regional Office for Europe, Moscow 125009, Russia; rodriguesdasilvabred@who.int; 4Laboratory of Dietetics and Quality of Life, Department of Food Science and Human Nutrition, School of Food and Nutritional Sciences, Agricultural University of Athens, 118 55 Athens, Greece; lpoulia@gmail.com

**Keywords:** caffeine, coffee, COVID-19, energy drinks, lifestyle, lockdown

## Abstract

The Coronavirus disease 2019 (COVID-19) has been characterized by the World Health Organization as a pandemic in March 2020 and the lockdown measures that were implemented in an effort to limit the transmission of the virus affected the daily life of many people in all over the world. The aim of this systematic review was to investigate the changes during/after the lockdowns in caffeine consumption by coffee and energy drinks. A systematic literature search was conducted in three databases (PubMed, Embase, Web of Science) up to 31 December 2021 and out of 19,511 studies found and 12,885 screened, 16 studies were included according to eligibility criteria. Results regarding coffee consumption showed that a significant part of individuals decreased their consumption and in five studies an increase was reported, including women and seniors >60 years old. Energy drinks were also consumed less during the lockdown compared to the pre-lockdown time. Attention should be given for menopausal women where an increase in coffee consumption was found which could impair bone density, but further research is needed in order to make safe conclusions.

## 1. Introduction

In March 2020, the Coronavirus disease 2019 (COVID-19) was declared by the World Health Organization as a pandemic [1]. As of 7 January 2022 more than 300,000,000 COVID-19 cases were reported and more than 5,400,000 people died from this disease. [2] Due to the severity of the acute respiratory syndrome (ARDS), which consists a severe complication of COVID-19 [3], along with the significantly rapid spreading of the virus, the governments of a great majority of countries around the world decided to implement protective measures in order to effectively slow down the transmission. With the first global lockdown being on January 2020 [4], a new reality of isolation and social-distancing was a fact which undoubtedly affected people’s daily life almost worldwide. Schools, universities, business’ and undoubtedly the whole socioeconomic structure had been affected, resulting to additional limitations in daily routine [4]. Simultaneously, depressive, anxiety and post-traumatic symptoms either in home-isolated individuals or in hospitalized COVID-19 patients were reported [5].

The impact of the measures taken to prevent the spread, could also be reflected on lifestyle-related habits and patterns which were found to be affected in many countries [6,7,8,9,10,11]. According to the literature, drinking coffee or other caffeine-containing drinks can have positive effects on mood and reduction of stress and can also enhance attention and alertness [12,13,14]. Therefore, a change in lifestyle behaviors such as coffee and energy drinks consumption that can also boost performance and endurance [13], could be a way to respond to the pandemic-related stress.

Changes in caffeine consumption during the period of self-isolation and/or lockdowns, are important to be identified, as they could be related with both physical and mental health consequences [12,13,14]. According to our knowledge there is no published study in which the behavior pattern of caffeine consumption during the pandemic was examined. In this systematic review we aimed to investigate the impact of lockdowns on the consumption of the main sources of caffeine (i.e., coffee and energy drinks).

## 2. Materials and Methods

This is a systematic review which has been conducted according to the Preferred Reporting Items for Systematic Reviews and Meta Analyses (PRISMA) guidelines [15] (Appendix A). The protocol of this study in which the research methodology of the study is described in detail was submitted to the OSF platform (https://osf.io/2mye7/, accessed on 3 December 2021). 

### 2.1. Search Strategy

A systematic literature search in the largest and most up-to-dated databases, PubMed via MEDLINE, Web of Science and EMBASE has been conducted independently by two authors (DB and KE), up to 31 December 2021, using the following search string for PubMed database: “((caffeine) OR (coffee) OR (energy drinks) OR (nutrition) OR (eating habits) OR (lifestyle)) AND ((COVID-19) OR (COVID) OR (COVID-19) OR (SARS-CoV-2) OR (lockdown))”, and it has been modified accordingly for the other databases. (See Appendix A for details). The reference lists of the articles selected were added in a reference manager software (Endnote X9 for Windows, Thomson Reuters) and were screened manually in order to obtain additional data. Subsequently, total hits obtained from the three databases were pooled together and duplicate articles were removed. Retrieved articles were screened by reading the article ‘title’, ‘abstract’ and ‘full-text’. The filtered articles were further screened by reading the individual manuscripts, and those not satisfying inclusion criteria (given below) were excluded. After duplicates removal, the remaining studies were screened for eligibility by two reviewers (DB and KE). Any disagreement was solved by the involvement of a third reviewer (MC). Reference lists of the retrieved articles were also screened for other relevant articles. 

### 2.2. Inclusion-Exclusion Criteria

In this systematic review we searched for observational studies (prospective, retrospective cohort studies and cross-sectional studies) in which information regarding caffeine consumption before and during/after the lockdown period was reported. Eligibility criteria were the following: (1) adult population (≥18 years old), (2) reported caffeine consumption (number/portion/quantity of coffee and/or energy drinks, or grams of caffeine), (3) Information regarding change in any of caffeine, coffee, or energy drinks consumption during/after the lockdowns period. Studies reporting tea consumption were excluded, due to the numerous variations of tea and the uncertainty regarding the levels of caffeine that is contains. Editorials, reviews, and studies published not either in English or Spanish language were also excluded.

### 2.3. Data Extraction-Quality Assessment

Data from included studies were extracted independently by two reviewers (DB and KE) in a standardized Microsoft Excel^®^ form. Any disagreement was solved by consensus. Information extracted from each study were country origin, sex, age, and the subgroup of participants if applicable, time of the survey conducted and information regarding caffeine, coffee, and energy drinks consumption before and during/after the lockdown periods. In case of missing data, the corresponding authors were to be contacted by email. Main outcome of this study was to examine the change on caffeine consumption before and during/after the lockdowns by examining its main sources, namely coffee and energy drinks consumption. 

Assessment of the quality of included studies was performed by two authors (DB and KE) using the Joanna Briggs Institute (JBI) Critical Appraisal tools for cross sectional studies [16]. Any discrepancy was solved by consensus.

## 3. Results

### 3.1. Search Results

In total, 19,511 studies were identified though the systematic literature search. After removing duplicates, 12,885 were screened for eligibility. Finally, according to our inclusion/exclusion criteria, 16 studies were characterized as acceptable to be included in this systematic review [17,18,19,20,21,22,23,24,25,26,27,28,29,30,31,32]. During full-text articles assessment, studies which were not relevant to the review topic, in which caffeine consumption before and after/during lockdown was not reported (wrong outcome), which did not examine adults (wrong population) and which were not observational studies (wrong study design), were excluded. All the details regarding the eligibility process of the included studies can be seen in the flow diagram in Figure 1. 

Characteristics of the eligible studies can be found in Table 1 and Table 2 including country origin, age, sex and the subgroup of population if applicable. In total 67,241 adults from Austria [31], Colombia [29], Croatia [23], China [26], France [21], Germany [28], Italy [19,20,22,27], Kuwait [25], Poland [18,24,31], Saudi Arabia [17], Turkey [32], and USA [30] were included.

### 3.2. Quality Assessment

Results of the quality assessment according to the JBI Checklist can be found in Figure 2 and Table 3. According to the judgment of the authors, two out of 16 studies were characterized as “best” regarding their quality [18,21]. Regarding the remaining 14 studies unclear statements and/or biases were detected with regards to questions referred to validation of exposure measures, criteria for measurement, identification of confounding factors and strategies to deal with [19,20,22,23,24,25,26,27,28,29,30,31,32]. 

### 3.3. Coffee Consumption

In 14 out of 16 studies information about coffee consumption was reported [17,19,20,21,22,23,25,26,27,28,29,30,31,32]. In the majority of included studies, coffee consumption tended to stay stable during/after the lockdown periods from February to December 2020 [19,20,22,23,26,27,28,32]. In one study from Saudi Arabia [17], despite the slight decrease of the amount of caffeine that was reported during/after the lockdown period (11.3 g/d during/after vs. 11.6 g/d before the lockdown), the overall quantity of coffee (in mL) was found to be increased during/after the lockdown period compared to the pre-lockdown time (159.4 mL/d vs. 143 mL/d). Increased coffee consumption was reported in France, by Deschasaux-Tanguy [21], where 13.5% of the participants stated that their coffee consumption was higher during/after the lockdown compared to the pre-lockdown period. However, in the same study information regarding individuals who did not change their consumption is missing. In a group of menopausal women from Italy coffee consumption was increased for a significant part of the participants (28.6 and 29.0%) during the two lockdown periods [20]. Similarly, increased consumption was reported for almost a third of participants in studies led by Palmer and Yilmaz [28,32]. Moreover, servings of coffee were found to be slightly increased in a Colombian study led by Pertuz-Cruz (1.61 servings after/during the lockdown compared to 1.57 servings in the pre-lockdown time) [29].

In the study led by Husain regarding the Americano coffee, [25] subjects who consumed moderate amounts of coffee stated that their consumption was decreased during the lockdown period, whereas subjects who consumed more than 5 cups of coffee per day, increased their coffee consumption during/after the lockdown. Arabic coffee consumption was also decreased for moderate coffee drinkers and only for adults that used to drink more than 6 cups of coffee per day, a slight increase in their consumption was reported (form 4.6% before to 4.8% after/during the lockdown) [25]. Details regarding coffee consumption before and after/during the lockdown as were found in eligible studies are presented in Table 2. 

Decreased caffeine consumption after/during the lockdown was stated in a study from USA (247 mg before and 234 mg after/during the lockdown) [30] and in the same pattern, coffee consumption was reported as decreased after/during the lockdown with regards to Polish, Austrian and UK populations [31].

### 3.4. Energy Drinks 

Information with regards to energy drinks consumption was stated only in two studies, both from Poland (18, 24). Energy drinks consumption was found to be unchanged for most of the participants (93.4%) according to Górnicka et al., whereas 5% of the participants decreased their consumption [24]. Moreover, in Błaszczyk-Bebenek et al. [18] the frequency of energy drinks consumption was found to be decreased after/during the lockdown period, with a slight increase (from 0.3–0.6%) reported among individuals who consumed these beverages once a day. Results of energy drinks consumption before and after/during the lockdown period can be found in Table 3.

## 4. Discussion

According to our knowledge this is the first systematic review that aimed to investigate the changes in caffeine consumption during/after the lockdowns. More than 67,000 adults from a significant part of the world were examined regarding their caffeine, coffee and energy drinks consumption. 

Results with regards to coffee consumption showed that for the majority of individuals this habit was not changed. However, for a significant part (8.4–26.4%) of individuals this consumption was decreased [17,19,21,22,26,27], and the highest reduction was observed in Italy [27]. The most possible explanation for this decrease might be the closing of coffee shops during the period of lockdown and the significant less opportunities to meet with friends “for a coffee”. In three studies from Italy where different groups of people were examined, coffee consumption presented a tendency towards decrease for the general population [27] and seniors >60 years old [22], but a tendency towards increase was reported for menopausal women in all the lockdowns (20). Decreased consumption was also found in data from Austria, Poland, UK and USA [30,31]. Increased consumption was reported in a three studies where general population was included [21,28,29,32]. In a great interest is the fact that in the study from Saudi Arabia quantity of coffee was found to be increased but the amount of caffeine containing was lower compared to the pre-lock down time [17]. One possible explanation could be that during the restriction time people continued to consume habitually coffee, but they chose to avoid the effects of caffeine by using decaffeinated products. Nevertheless, further research is necessary in this field. 

There is little evidence regarding moderate consumption of coffee and health risks [33,34]. Some data support that caffeine has positive and/or negative impact on brain [35,36,37,38,39], lungs [40,41], liver [42,43,44], heart [34,45,46] and endocrine [47,48] function or acts as immunomodulator [49] but further research is needed in the field of clinical impact of coffee. 

However, of great interest is the fact that a tendency towards an increase of coffee consumption was found in the subgroup of menopausal women [20]. Caffeine consumption even by consuming one cup of coffee per day, can increase the risk of fractures especially for women [34,50]. During the menopause, estrogen deficiency impairs the bone density [51] and therefore caffeine consumption should be controlled and limited. Moreover, caffeine has been blamed among factors that enhance menopausal symptoms and particularly vasomotor symptoms [52]. 

In addition to that, increased caffeine intake has been linked with adverse effects on aquatic ecosystems which could be a dangerous pollutant [53]. Moreover, according to the European Food Society Authority (EFSA), coffee consumption leads to increased exposure of furan and other compounds, such as 2- and 3- methylfurans, which may be responsible for liver impairments and neoplasms [54]. Acrylamide, which is known for its carcinogenic effects, is also found in coffee products [55], and roasted coffee is major source of ochratoxin A which consists a health concern due to its genotoxicity [56].

On the other hand, there is evidence which support that caffeine consumption can reduce the incidence of post-menopausal breast [57] and endometrial cancer [58]. In general, according to recent studies caffeine can be a part of a diet and some negative outcomes that emerge from coffee consumption may not a result of caffeine itself [59,60]. 

In the Husain et al. study, which had data about the frequency of consumption of Americano and Arabic coffee, a decrease in consumption was reported for both types of coffee [25]. The extent of decrease of Americano coffee was found to be greater in comparison to Arabic coffee reduction possibly due to closed coffee shops [25]. However, not only Arabic coffee but also other types of coffee have been associated with increased total cholesterol [61], triglycerides and LDL levels [62] and therefore might deteriorate the lipidemic profile of coffee drinkers. 

Regarding energy drinks only in two studies from Poland relevant data was available [18,24]. Energy drinks usually contain 80–150 mg per 236 mL [63], and they can lead to cardiovascular [64,65,66], neurological [67,68,69] and gastrointestinal [70,71] negative effects. Even though alcohol consumption showed a tendency towards increase during the COVID-19 lockdowns [10], energy drinks which often consumed in combination with alcohol, did not follow the same pattern and possible explanations might be the closing of nightclubs and lower need for an “energy boost” during the lockdown periods. 

This study can be characterized by several strengths. Firstly, this is the first systematic review that investigated caffeine consumption after/during the lockdown periods by examining coffee and energy drinks consumption. Eleven studies were included from a significant geographical part of the world and more than 67,000 adults were included meaning that a quite large sample of population has been examined. Moreover, the quality of studies, according to the JBI checklist, was rather satisfactory making our results accurate enough. Limitations of our studies include the fact that information regarding the representativeness of each sample size was not reported in all of them. In addition, results for energy drinks consumption were only from one country. Besides, the fact that information of the studies was provided by self-reported questionnaires could lead to biases. Moreover, *p*-values were not reported in all studies and information regarding statistical significance is missing. Furthermore, the absence of evidence regarding the content of caffeine and the missing information in cases of decaffeinated coffee consumption could affect the accuracy of our results. Lastly, only studies in English and Spanish language are part of this systematic review and therefore related articles in any other language are not part of this study. Due to the lack of data, we were not able to perform a meta-analysis.

## 5. Conclusions

Lockdowns undoubtedly affected lifestyle behaviors of adults. Caffeine consumption showed a tendency towards decreased for a significant part of people. Attention may be needed regarding coffee consumption for menopausal women, but further research is needed to make safe conclusions. Energy drinks consumption was found to be decreased which is considered to be a positive impact of the lockdown period. 

## Figures and Tables

**Figure 1 ijerph-19-05255-f001:**
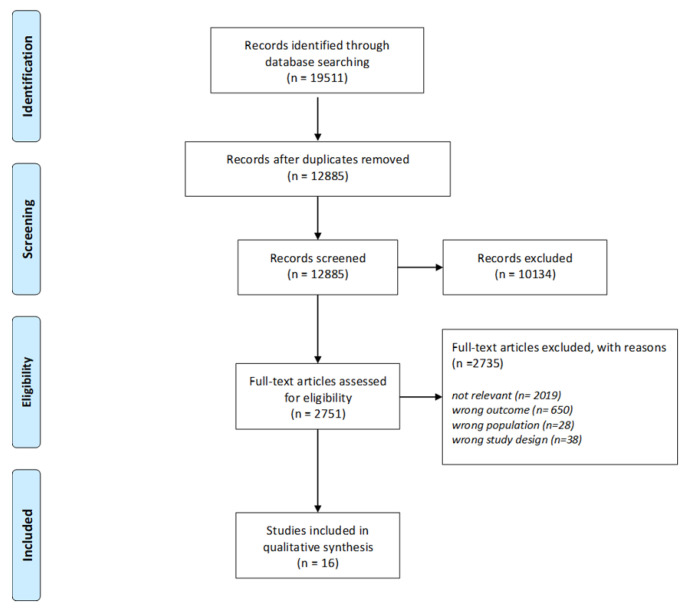
Flow diagram of the study selection process.

**Figure 2 ijerph-19-05255-f002:**
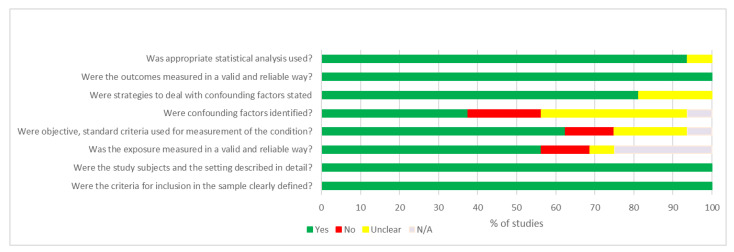
Items of JBI Critical Appraisal Checklist of included studies.

**Table 1 ijerph-19-05255-t001:** Characteristics of Included studies for coffee consumption and relevant info before and after/during lockdowns.

First Author, Year (Country)	Individuals (F/M/O) Specific Subgroup of Included Population	Age	Time of Survey Conduction	Coffee before Lockdown	Coffee After/during Lockdown	*p* Value	Coffee Increased	Coffee Decreased	Coffee No Change
Al Musaraf, 2021 (Saudi Arabia)	297 (297/0/0)	19–30	February 2019–May 2020	11.6 (17 ) * g/d 143 (199) * mL/d	11.3 (19) * g/d 159.4 (243) * mL/d	0.87 0.35			
Cirilo, 2021 (Italy)	140 (140/0/0) Infertile women	18–49 /39.4 (5) *	20 April–4 May 2020				5%	20.30%	74.70%
Coppi, 2021 (Italy)	320 (320/0/0) Menopausal women	45–54	1st: 24 March–3 May 2020 & 2nd: 10–20 December 2020				1st: 28.6% 2nd: 29%	1st: 18% 2nd: 16%	1st: 53.4% 2nd: 55%
Deschasaux-Tanguy, 2021 (France)	37,252 (19,483/17,769/0) General population	52.1 (16.6) *	April–May 2020				13.5%	8.40%	
Di Santo, 2020 (Italy)	126 (102/24/0) Seniors >60 years of age with mild cognitive impairment or subjective cognitive decline	74.29 (6.51) *	21 April–7 May 2020				6.60%	8.50%	84.9%
Dogas, 2020 (Croatia)	3027 (1989/506/0)	40 (30–50) ^#^	25 April–5 May 2020	2.1 (1.0) cups/d	2.1 (1.1) cups/d	0.003			
Husain, 2020 (Kuwait)	415 (285/130/0)	18–73/38.47 12.73) *	30 March–15 April 2020	Americano coffee: None: 22.7% <1: 24.1% 1–2/d: 41.4% 3–4/d: 8.7% 5–6/d: 1.9% >6/d: 1.2% Arabic coffee: None: 46.3% <1: 17.3% 1–2/d: 15.4% 3–4/d: 10.6% 5–6/d: 5.8% > 6/d: 4.6%	Americano coffee: None: 31.6% <1: 25.5% 1–2/d: 33% 3–4/d: 5.5% 5–6/d: 2.9% >6/d: 1.4% Arabic coffee: None: 54.7% <1: 13.7% 1–2/d: 13.5% 3–4/d: 7. % 5–6/d: 5.5% >6/d: 4.8%				
Jia, 2021 (China)	10,082	15–28/19.8 (2.3) *	9–12 May 2020				2.50%	16%	Never: 96.4% Constant: 12.65%
Maffoni, 2021 (Italy)	1304 (973/331/0)		30 April–10 May 2020				7.9%	26.4%	65.7%
Palmer 2021 (Germany)	827 (622/205)	>18	12 March–3 May 2020				31.9%	8.4%	59.7%
Pertuz-Cruz 2021 (Colombia)	11,490 (4012/7478)	>18	6 April and 22 May 2020	1.57 ^	1.61 ^				
Shaw 2021 (USA)	24 (15/10)	37.6 (9.3) *	May to mid-June 2020	247 (160) * mg	234 (128) * mg				
Skotnicka. 2021 (Poland, Austria)	1071 (604/467)	>18	1 October to 30 October 2020	Poland: at least once a day: 76.9% Austria: 62.61% UK: 54.34%	Poland: at least once a day: 76.2% Austria: 62.32% UK: 49.84%				
Yilmaz 2020 (Turkey)	866 (677/189)	21.2 (1.4) *	5–6 April 2020				30.6%	6.7%	62.7%

* Mean (SD), ^#^ Median (IQR). F: Female, M: Male, O: Other. ^ (servings).

**Table 2 ijerph-19-05255-t002:** Characteristics of included studies for energy drinks consumption and relevant info before and after/during lockdowns.

First Author, Year (Country)	Subjects (F/M/O)	Age	Time of Survey Conducting	Energy Drinks before Lockdown	Energy Drinks after/during Lockdown	*p* Value	Energy Drinks Increased	Energy Drinks Decreased	Energy Drinks No Change
Błaszczyk-Bebenek, 2020 (Poland)	312 (200/112)	41.12 (12.04) *	29 April–19 May	Never: 78.5 % 1–3/month: 14.7% Once a week: 1.6% Few times a week: 4.2% Once a day: 0.3% Few times a day: 0.6%	Never: 85.3% 1–3/month: 8.3% Once a week: 3.2% Few times a week: 1.9% Once a day: 0.6% Few times a day: 0.6%	0.015			
Górnicka, 2020 (Poland)	1381 (2138/243)	>18	30 April–23 May 2020				1.60%	5%	93.40%

* Mean (SD). F: Female, M: Male, O: Other.

**Table 3 ijerph-19-05255-t003:** Results of the quality assessment of included studies according to the JBI Checklist.

First Author, Year	Were the Criteria for Inclusion in the Sample Clearly Defined?	Were the Study Subjects and the Setting Described in Detail?	Was the Exposure Measured in a Valid and Reliable Way?	Were Objective, Standard Criteria Used for Measurement of the Condition?	Were Confounding Factors Identified?	Were Strategies to Deal with Confounding Factors Stated?	Were the Outcomes Measured in a Valid and Reliable Way?	Was Appropriate Statistical Analysis Used?
Al Musharaf, 2021								
Blaszczyk-Bebenek, 2021								
Cirilo, 2021								
Coppi, 2021								
Deschaseaux-Tanguy, 2021								
Di Santo, 2020								
Dogas, 2020								
Górnica, 2020								
Husain, 2020								
Jia, 2020								
Maffoni, 2021								
Palmer, 2021								
Pertuz-Cruz, 2021								
Shaw, 2021								
Skotnicka, 2021								
Yilmaz, 2020								

Green: YES, Red: NO, Yellow: Unclear, White: Not applicable.

## Data Availability

Not applicable.

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
