# Peer review of "The Impact of Lockdowns on Caffeine Consumption: A Systematic Review of the Evidence"

_ijerph, 2022, doi:10.3390/ijerph19095255_

Round 1
Reviewer 1 Report
Dear Authors,
The study was structured very carefully, although it contains many subjective elements that are difficult to verify.
The starting points for the study and the assumptions should be clearly stated in the „Introduction” part.
„Wrong outcome”? What does it mean? (In Figure 1, „Eligibility” part) It is not clear, please give explanation!
In Figure 2 the fifth row is unreadable, which is disturbing and the „source” is missing.
Explanations needed for the selection of the database.
In Row 125: „Results of the quality assessment according to the JBI Checklist can be found in Figure 2 and Supplementary Table 2.” Table 2 contains other information.
In Row 127: „Regarding the remaining 14 studies unclear statements and/or biases were detected with regards to questions referred to validation of exposure measures, criteria for measurement, identification of confounding factors and strategies to deal with.” à Can you give examples for these biases?
Row 132: „In 15 out of 17 studies information about coffee consumption was reported” There is a mistake, because 16 instead of 17?
Row 228-229 „Eleven studies were included from a significant geographical part of the world and more than 67000 adults were examined leading to a quite representative sample.” Why you think it is representative? I think 16 studies from 11 countries were included.
All in all, it is very interesting and provides a good overview of the studies available on the subject.
The relationship between a greater shift towards a healthy lifestyle and coffee consumption should be built in and also further analyses in this field are needed.
Reviewer 2 Report
Bakaloudi et al. produced an interesting systematic review on the impact of lockdown on caffeine consumption, extracting and considering 16 evidence in the literature. The manuscript appears written with scientific rigor and it allows for a fluid reading.
However, further efforts must be aimed at improving the manuscript, among which:
- Does the review concern the consumption of "caffeine" or "coffee"? In all the selected articles was the quantity of coffee consumed by the investigated population clearly indicated (having been quantified and measured)? For example, an Italian espresso coffee is very different from an American coffee, in terms of caffeine content. Please, better detail this aspect;
- do the studies considered make clear the presence of any bias regarding data relating to the number of coffees daily consumed that were self-reported by the interviewees?
- In discussions section, please consider whether the use of domestic automatic machines (eg via capsules or pods) could also have impacted the results. If this data does not emerge, it would be interesting to understand if the number of automatic machines sold online and of capsules / pods has increased starting from January 2020 (there are market analyzes in this regard).
- LL 212-222: please also consider the possible presence of contaminants, including heavy metals, mycotoxins and acrylamide, to make the article attractive on a Public Health topic.
Finally, I also believe it is essential that the Authors respect the provisions of the Instructions for the Authors of IJERPH, in particular: to indicate the affiliations of the Authors, using the superscript numbers and not the letters; indicate in the text all the references using square brackets [] and not round (); in the References section please indicate the name of the journals using Journal Title Abbreviations; comply with the type of italic or bold font where requested in the aforementioned instructions; provide further details regarding citations numbers 2 and 16. The work, moreover, requires partial revision in English, both for the form and for the syntax.
Thank you for your efforts in perfecting this systematic review.
Round 2
Reviewer 2 Report
Thank you for your commitment and for responding adequately and quickly to almost all questions, providing clarifications and making changes and additions to the paper.
Regarding "Point 5" (LL 212-222: please also consider the possible presence of contaminants, including heavy metals, mycotoxins and acrylamide, to make the article attractive on a Public Health topic), the answer provided is not exhaustive, since that I would consider it appropriate to clarify the main chemical contaminants of this specific food matrix, also evaluated by EFSA ( https://www.efsa.europa.eu/en/press/news/furan-food-efsa-confirms-health-concerns - https://www.efsa.europa.eu/en/topics/topic/acrylamide - https://efsa.onlinelibrary.wiley.com/doi/full/10.2903/j.efsa.2020.6113 -). Considering the relevance of the Journal, I therefore recommend the inclusion of a short paragraph in the discussions relating to what was previously requested, also to make the paper more attractive to the professional specialist in this sector. Thank you.
